# Implementation of field detection devices for antimalarial quality screening in Lao PDR—A cost-effectiveness analysis

Nantasit Luangasanatip[1]*, Panarasri Khonputsa[1], Céline Caillet[2,3,4], Serena Vickers[2,3,4], Stephen Zambrzycki[5], Facundo M. Fernández[5], Paul N. Newton[2,3,4], Yoel Lubell[1,3]

**1** Mahidol-Oxford Tropical Medicine Research Unit, Faculty of Tropical Medicine, Mahidol University, Bangkok, Thailand, **2** Lao-Oxford Mahosot Hospital Wellcome Trust Research Unit, Microbiology laboratory, Mahosot Hospital, Vientiane, Lao PDR, **3** Centre for Tropical Medicine and Global Health, Nuffield Department of Clinical Medicine, University of Oxford, Oxford, United Kingdom, **4** Infectious Diseases Data Observatory (IDDO)/Worldwide Antimalarial Resistance Network (WWARN), University of Oxford, Oxford, United Kingdom, **5** School of Chemistry and Biochemistry, Georgia Institute of Technology, Atlanta, Georgia, United States of America

* nantasit@tropmedres.ac

**Data Availability Statement:** All relevant data are within the manuscript and its Supporting Information files.

## Abstract

Substandard and falsified (SF) antimalarials have devastating consequences including increased morbidity, mortality and economic losses. Portable medicine quality screening devices are increasingly available, but whether their use for the detection of SF antimalarials is cost-effective is not known. We evaluated the cost-effectiveness of introducing such devices in post-market surveillance in pharmacies in Laos, conservatively focusing on their outcome in detecting SF artemisinin-based combination therapies (ACTs). We simulated the deployment of six portable screening devices: two handheld near-infrared [MicroPHA-ZIR RX, NIR-S-G1], two handheld Raman [Progeny, TruScan RM]; one portable mid-infrared [4500a FTIR] spectrometers, and single-use disposable paper analytical devices [PADs]. We considered two scenarios with high and low levels of SF ACTs. Different sampling strategies in which medicine inspectors would test 1, 2, or 3 sample(s) of each brand of ACT were evaluated. Costs of inspection including device procurement, inspector time, reagents, reference testing, and replacement with genuine ACTs were estimated. Outcomes were measured as disability adjusted life years (DALYs) and incremental cost-effectiveness ratios were estimated for each device compared with a baseline of visual inspections alone. In the scenario with high levels of SF ACTs, all devices were cost-effective with a 1-sample strategy. In the scenario of low levels of SF ACTs, only four devices (MicroPHAZIR RX, 4500a FTIR, NIR-S-G1, and PADs) were cost-effective with a 1-sample strategy. In the multi-way comparative analysis, in both scenarios the NIR-S-G1 testing 2 samples was the most cost-effective option. Routine inspection of ACT quality using portable screening devices is likely to be cost-effective in the Laos context. This work should encourage policy-makers or regulators to further investigate investment in portable screening devices to detect SF medicines and reduce their associated undesired health and economic burdens.

**Funding:** This work has been co-funded by the Regional Malaria and other Communicable Disease Threats Trust Fund, which has been co-financed by the Government of Australia (Department of Foreign Affairs and Trade); the Government of Canada (Department of Foreign Affairs, Trade and Development); and the Government of the United Kingdom (Department for International Development). The grant (RETA 8763) was managed by the Asian Development Bank (https:// www.adb.org/) and awarded to PNN. Additional support was provided by Wellcome Trust Grant N° 202935/Z/16/Z (https://wellcome.org/). The funders had no role in study design, data collection and analysis, decision to publish, or preparation of the manuscript.

**Competing interests:** The authors have declared that no competing interests exist.

## Author summary

Distribution of poor quality medicines are an increasing global concern, especially in low- and middle-income countries (LMICs) where the effectiveness of antimicrobials can be a matter of life-or-death for patients with malaria and other potentially fatal infectious diseases. Substandard and falsified antimalarial drugs, including artemisinin-based combination therapies are widely distributed across LMICs. This endangers patients and in the longer term threatens malaria control and elimination campaigns by promoting the development of drug resistance. New field detection devices are increasingly available and could enhance the inspection process with prompt and actionable, real-time results. We conducted a cost-effectiveness analysis of the implementation of six portable devices for medicine quality screening during pharmacy post-market surveillances in Laos. This analysis conservatively focused only on the benefits of these devices in detecting substandard and falsified artemisinin-based combination therapies (ACTs), measured in terms of health outcomes for malaria patients obtaining treatment from pharmacies. Our findings suggest that using these portable devices for routine surveillance of ACT quality is likely to be cost-effective. Policy-makers and regulators might therefore consider investment in these field detection devices to minimise undesired health and economic burdens associated with substandard and falsified medicines.

## Introduction

Substandard and falsified (SF) medicines have devastating health and economic implications [1,2]. Falsified medicines are medical products that are produced to deliberately and fraudulently misrepresent their identity, composition or source, while substandard medicines are those that are produced by authorised manufacturers but fail to meet quality standards or specifications [3,4]. The distribution of SF medicines is an increasing global concern [5–7]. The World Health Organization (WHO) estimated that in low- and middle-income countries (LMICs), approximately 10% of medicines are substandard or falsified. Reasons why LMICs are highly affected include poor governance, weak technical capacity and poor supply chain management [2,8]. The economic impact of SF medicines in LMICs has been estimated at US $30.5 billion per year [1].

Despite declines in transmission, malaria is still ranked fourth in amongst infectious diseases in terms of its global burden, following diarrheal diseases, HIV, and tuberculosis, respectively [9]. The incidence of malaria cases and deaths in Laos were reported as 10.2 cases per 1,000 population and less than 50 deaths in 2015 [10]. WHO reported the number of confirmed malaria cases in Laos in 2015 as 36,056 [10]. Malaria is curable but in the absence of effective treatment can be fatal. Artemisinin-based combination therapies (ACT) are now recommended as first-line therapy for *Plasmodium falciparum* malaria worldwide [11]. A recent systematic review suggested that SF antimalarial drugs, including ACTs, are widely distributed across LMICs with a mean prevalence of 19.1% [12]. The distribution of SF ACTs endangers patients and in the longer term can threaten malaria control and elimination campaigns by promoting the development of drug resistance [13–15].

In a national survey conducted in Laos in 2003, 22 out of 25 pharmacies across the country were found to supply falsified artesunate [16]. In a repeat survey with similar methodology in 2012, a quarter of ACT samples tested were outside the 90–110% pharmacopeial limits claimed on the label, but none of the samples were classed as falsified [17]. While routine pharmacy

post-market surveillances are performed by regulatory bodies to detect SF medicines, these rely on an initial screening by visual inspection, followed by a longer and expensive process for centralized pharmacopeial testing.

New field detection devices are increasingly available and could enhance the surveillance process with prompt and actionable results at the point of inspection. These devices vary widely in cost and underlying technology, with varying advantages and disadvantages when used in the field [18].

This is the fourth paper in the Collection 'A multi-phase evaluation of portable screening devices to assess medicines quality for national Medicines Regulatory Authorities' [19]. Six devices deemed 'field-suitable' in the laboratory evaluation were included in a field evaluation in the Lao PDR (Laos) during which their utility and usability for pharmacy inspections were investigated [20]. This study aims to evaluate the costs and consequences of introducing these six portable devices for the screening of antimalarials during pharmacy inspections and assess whether investment in these devices would be cost-effective from a healthcare provider's perspective in the Laos setting. This analysis conservatively focuses only on the cost-effectiveness of these devices in detecting substandard and falsified ACTs, with health gains measured in terms of clinical outcomes for malaria patients obtaining treatment from pharmacies that undergo routine post-market surveillance.

## Methods

### Strategies

Six field detection devices deemed suitable based on laboratory evaluation and field-testing in Laos [21] were included in this analysis. These consist of two handheld near-infrared spectrometers, two handheld Raman spectrometers, one portable mid-infrared device, and single-use disposable paper analytical devices (PADs). All devices were compared with a baseline of visual inspections alone, that was assumed to detect 25% of substandard and 50% of falsified ACTs. This analysis estimated the incremental costs of the six devices if incorporated within these inspection visits, and their outcomes measured in disability adjusted life years (DALYs) averted, taking a healthcare provider's perspectives with a one year timeframe without discounting. A key assumption is that the introduction of the devices facilitating on-site detection of SF medicines will allow for the temporary removal of SF medicines from distribution in the pharmacies where they are detected.

The actual prevalence of SF ACTs in Laos is not well described, although the available evidence indicates a large decline in recent years in the prevalence of falsified antimalarials and modest falls in the prevalence of substandard antimalarials [17]. We therefore modelled two hypothetical scenarios with varying levels of SF ACTs in circulation. In Scenario 1, 60% of ACTs are genuine, 20% substandard, and 20% falsified. In Scenario 2, 85% are genuine, 10% substandard, and 5% falsified.

Almost all falciparum malaria cases in Laos are concentrated in 42 districts across five southern provinces (Savannakhet, Salavan, Sekong, Champasak, and Attapeu). Patients are assumed to be equally distributed across the districts and have equal access to 10 pharmacies per district. The key intervention to reduce the burden of malaria is the provision of long-lasting insecticide nets [22].

### Model structure

A decision tree model was developed to simulate post-market surveillance scenarios at the pharmacy level with screening devices as compared with visual inspection alone. Each pharmacy was assumed to stock three ACT brands which are used with equal frequency amongst

A

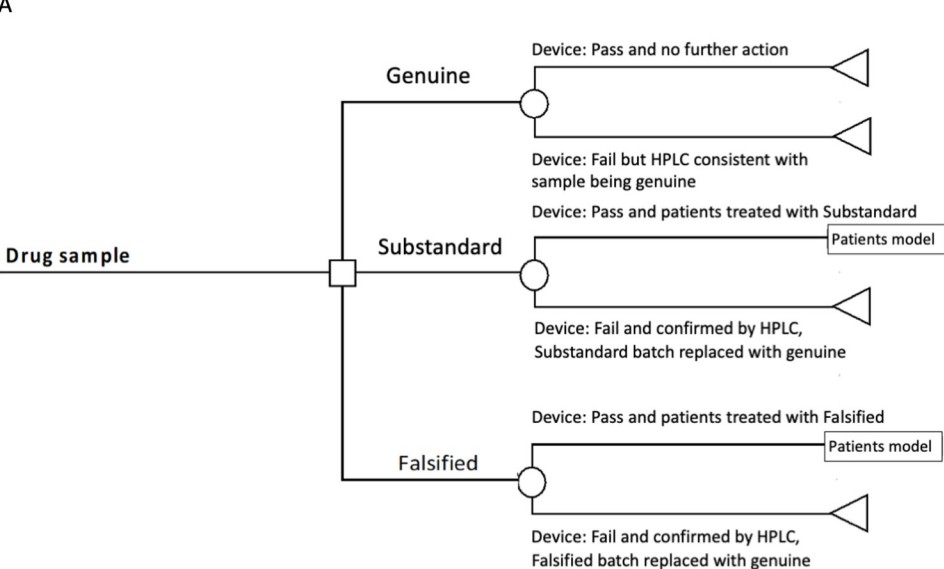

B

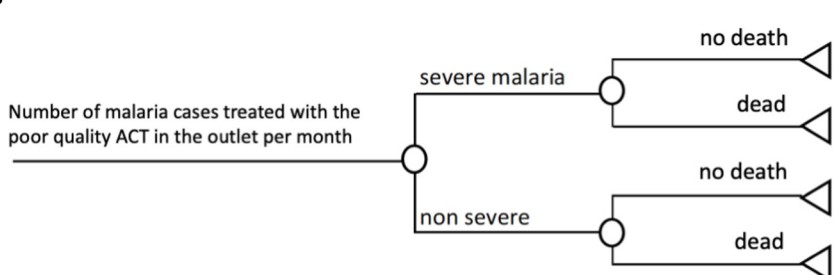

**Fig 1. Decision analytic model for drug sampling at pharmacy (A Medicines Model) and malarial illness (B Patients Model).**

malaria patients obtaining treatment from the pharmacy. The first component of the decision tree simulates the inspection process where three ACT brands are screened by inspectors in each pharmacy (Fig 1A). The second component models health outcomes for malaria patients prescribed an ACT, which can be genuine, substandard, or falsified (Fig 1B).

The modelled scenarios assumed that one device is available for each of the 42 malaria endemic districts for biannual post-market surveillances of 10 pharmacies per district. In each pharmacy and for each medicine, the inspectors test either one, two, or three samples per ACT brand. Testing higher numbers of samples implies a higher probability of the device identifying SF medicines, but also an increased probability of mistakenly indicating that a sample is not genuine. Performance estimates for the six devices were derived from their evaluation in a laboratory setting [19] determining the probabilities that they provide a correct result for either genuine, substandard or falsified antimalarials. The accuracy estimates were derived from the samples tested after removal from their packaging (Table 1). For sampling strategies in which two or three samples are tested, the probability of the device indicating a SF sample was raised to the power of the number of samples taken.

Where the screening devices indicates that a brand of ACT is SF, that batch is assumed to be replaced with genuine ACTs, implying a temporary improvement in the proportion of

**Table 1. Probability with which each device will identify genuine, substandard (between 50% and 80% API) and falsified (wrong/0% API) medicines with strategies of 1/2/3 samples per brand of ACTs.**

| Device | Medicine quality* | 1-sample | | 2- sample** | | 3-sample** | |
|---|---|---|---|---|---|---|---|
| | | *Device: Fail*[†] | *Device: Pass*[‡] | *Device: Fail*[†] | *Device: Pass*[‡] | *Device: Fail*[†] | *Device: Pass*[‡] |
| | Genuine | 0 | 1 | 0 | 1 | 0 | 1 |
| **TruScan RM** | Substandard | 0.42 | 0.58 | 0.66 | 0.34 | 0.80 | 0.20 |
| | Falsified | 1 | 0 | 1 | 0 | 1 | 0 |
| **MicroPHAZIR RX** | Genuine | 0 | 1 | 0 | 1 | 0 | 1 |
| | Substandard | 0.5 | 0.5 | 0.75 | 0.25 | 0.88 | 0.13 |
| | Falsified | 1 | 0 | 1 | 0 | 1 | 0 |
| | Genuine | 0 | 1 | 0 | 1 | 0 | 1 |
| **4500a FTIR** | Substandard | 0.33 | 0.67 | 0.56 | 0.44 | 0.70 | 0.30 |
| | Falsified | 1 | 0 | 1 | 0 | 1 | 0 |
| | Genuine | 0 | 1 | 0 | 1 | 0 | 1 |
| **Progeny** | Substandard | 0.08 | 0.92 | 0.16 | 0.84 | 0.23 | 0.77 |
| | Falsified | 1 | 0 | 1 | 0 | 1 | 0 |
| | Genuine | 0 | 1 | 0 | 1 | 0 | 1 |
| **NIR-S-G1** | Substandard | 0.33 | 0.67 | 0.56 | 0.44 | 0.70 | 0.30 |
| | Falsified | 0.95 | 0.05 | 1 | 0 | 1 | 0 |
| | Genuine | 0 | 1 | 0 | 1 | 0 | 1 |
| **PADs** | Substandard | 0 | 1 | 0 | 1 | 0 | 1 |
| | Falsified | 1 | 0 | 1 | 0 | 1 | 0 |

*Genuine medicine (90–110% API), Substandard (50% and 80% API) and Falsified medicine (wrong/0% API)

**The probability of detecting SF medicine under 2- and 3-sample strategies are estimated with each repeat sample test having the same probability as a single sample test independently.

[†]Device:Fail refers to the probability that the device indicates that the medicine is substandard or falsified.

[‡]Device:Pass refers to the probability that the device indicates that the medicine is genuine.

genuine medicines at the pharmacy. This was assumed to last for one month before returning to the scenario-specific baseline prevalence of SF ACTs. Samples that fail the device screening are assumed to be sent for formal reference laboratory testing by high-performance liquid chromatography (HPLC), a standardized and high cost analytical chemistry technique. Incorrectly classifying a genuine sample as a fail by the portable devices incurs the unnecessary costs of reference HPLC testing and replacement ACTs. If the device indicates a genuine medicine, no further action is taken. Therefore, if a sample classed as genuine was in fact substandard or falsified, patients remain at higher risk of severe outcomes.

## Parameter inputs

**Costs.** The total cost of introducing the screening devices was composed of costs for procurement and delivery, inspector visits, sample test reagents, reference testing by HPLC, and the cost of drug replacement if the results from a tested sample failed the screening test. A micro-costing approach was used to estimate these costs. To estimate the cost of devices and delivery, fixed and variable costs for the devices and their consumables were collected from either manufacturer responses to a list of questions sent by email or the supplier quotation (Text A in S1 Text). Fixed costs comprised of the instrument purchase and maintenance assuming a five-year shelf life for all devices, other than the PADs which are single-use disposable tests. Variable costs include those for reagents and materials used for each assay, as well as for the additional time spent per sample by inspectors.

For the costs of inspections, five inspectors per district were assumed to perform inspections at 10 pharmacies biannually. Based on informal interviews with staff from the Lao Bureau of Food and Drug Inspection, all five inspectors were assumed to visit two pharmacies per field trip. The total number of visits and hours spent for inspections was multiplied by the inspectors' salary and per diem rates to calculate the total cost per inspection. The cost of additional inspection time for use of the devices was derived from a time-and-motion study in a simulated pharmacy conducted with Lao medicine inspectors [19] (Table A in S1 Text). The total variable costs per inspection were determined by the sampling strategy of either one, two, or three samples per ACT brand for each of the three ACT brands. The costs of HPLC reference testing and ACT replacement were calculated assuming that all samples failing a device test were tested with HPLC, and SF stocks replaced with genuine ACTs.

**Probabilities and consequences.**   Patients treated with a substandard or falsified medicine are assumed to be at higher risk of becoming severely ill (24%) [27] and dying of malaria (15% of those severely ill) [23]. These adverse outcomes are converted into disability adjusted life years (DALYs), using the assumption that the event of illness due to severe malaria would last for six weeks with 0.133 disability weight [24] and the average age of patients infected with malaria was conservatively assumed to be 48 years. Therefore, those who died of malaria would lose 20 life-years, based on the average life expectancy in Laos [25]. (Table 2).

**Cost-effectiveness analysis.**   The costs and DALYs were used to estimate the incremental cost-effectiveness ratios (ICER) of each device in both scenarios of high and low prevalence of SF antimalarials. Devices are considered cost-effective when the incremental cost per DALY averted is below the assumed willingness to pay threshold (WTP) of US$ 2,353, the 2016 Laos GDP per capita [26], as recommended by the WHO [28]. ICERs were also estimated for different sampling strategies, whereby the inspectors select either one, two, or three samples per brand of ACT.

The ICERs were initially calculated for each device individually as compared with a baseline of visual inspections alone. To facilitate the comparison of multiple devices and sampling strategies we use the net-monetary benefit (NMB). The NMB provides a simpler indicator of relative cost-effectiveness than ICERs, whereby the device and sampling strategy with the highest NMB is identified as optimal. NMB is calculated by multiplying the effectiveness of the intervention (in this instance measured in DALYs averted) by the WTP threshold and deducting any incremental costs associated with use of the device.

A series of one-way sensitivity analyses were performed to determine the effect on results if parameter values deviated from the initial point estimates. For brevity, this was carried out only in the lower prevalence scenario. Plausible ranges for key parameters, including the cost of the devices (-50% and +20%), test performance (-30% and +30%), and number of years of life lost due to malaria (-20% and +20%), were applied to the model. The results are presented in a tornado diagram to show the magnitude of the effect of varying these parameter estimates on the cost-effectiveness of each device. In addition, an alternative approach of purchasing one device per province instead of one per district (i.e. purchasing five instead of 42 devices), was also evaluated; the costs of transporting the devices between districts was not accounted for. As the assumption regarding the detection rates of visual inspection for substandard and falsified are central to the analysis (25% and 50%, respectively), two further scenario analyses were performed with detection rates of 12.5% and 5% for substandard medicines, and 25% and 10% for falsified medicines.

Finally, a budget impact analysis estimating the financial impact when introducing these devices at the country level was also carried out. This consisted of the initial purchase costs for 42 devices (1 per district), and the annual costs for using the devices in the post market surveillances over a duration of 5 years.

**Table 2. List of parameters used in the cost-effectiveness analysis model.**

| Parameters | Values | Reference |
|---|---|---|
| Total malaria cases per year (Lao PDR, year 2015) | 36,056 | [10] |
| Number of districts where malaria cases were reported | 42 | |
| Number of pharmacies inspected per district | 10 | Laos Food and Drug Department (FDD)* (current practice) |
| Number of ACT brands per pharmacy | 3 | Assumed |
| Total number of malaria cases, per pharmacy per year | 86 | [10] |
| Total ACT (blisters) stock of all brands, per pharmacy | 258 | Assumed |
| Number of samples tested per brand | 1–3 | Assumed |
| Number of inspections, per pharmacy per year | 2 | Lao FDD |
| Number of months genuine replacement ACTs in place until returning to baseline levels | 1 | Assumed |
| **Economic data** | | |
| Number of inspectors, per visit (five inspectors per district to perform inspections at 10 pharmacies) | 5 | Laos FDD |
| Hours per inspection, per pharmacy | 1 | Assumed |
| Number of pharmacy visit, per day | 2 | Assumed |
| Inspector's salary per hour (US$ 144 or 1.2 million LAK per month) | 0.9 | Bureau of Food and Drug Inspection (BFDI), Lao PDR |
| Per diem (per day) (250,000 LAK) | 30 | Bureau of Food and Drug Inspection (BFDI), Lao PDR |
| Cost of device (up front and subsequent costs over 5 years) | See Table 3 | [20] |
| Cost of test, per sample (consumable material and reagents) | See Table 3 | [20] |
| Cost of reference analysis with HPLC (1,245 million LAK), per sample | US$ 149.4 | Bureau of Food and Drug Inspection (BFDI), Lao PDR |
| Cost of ACT, per tablet | US$ 0.78 | [23] |
| Years of life with disability (YLD) due to severe malaria | 0.02 | Disability weight of 0.133 [24] and an assumed duration of 6 weeks |
| Years of life lost (YLL) | 20 | Assumed based on life expectancy in Laos [25] and most patients in low endemic settings being adults |
| Willingness to pay (GDP per Capita) threshold (Lao) | US$ 2,353 | [26] |
| **Transition Probability** | | |
| Risk of severe malaria when treated with genuine ACT | 0 | [27] |
| Risk of severe malaria when treated with substandard and falsified ACT | 0.24 | [27] (Assumed to be equal to untreated malaria; average for children and adults) |
| Risk of death in severe malaria | 0.15 | [21] |
| Risk of death in non-severe malaria | 0 | [21] |

*MRA–Medicines Regulatory Authority

# Results

## Cost of screening device

The TruScan RM had the highest fixed cost, followed by Progeny, MicroPHAZIR RX, 4500a FTIR, NIR-S-G1, and PADs (Table 3). The largest proportion of the total costs for all devices

**Table 3. Fixed and variable costs of the devices.**

| Costs (US$, 2017) | TruScan RM | Micro PHAZIR RX | 4500a FTIR | Progeny | NIR-S-G1 | PADs |
|---|---|---|---|---|---|---|
| **Initial cost** | | | | | | |
| - Cost of devices* | 68,750 | 52,250 | 34,724 | 67,449 | 1,539 | 0 |
| **Subsequent cost (over 5 years)** | | | | | | |
| - Replacement cost of the battery | 112 | 506 | N/A | 580 | 30 | N/A |
| - Light bulb | N/A | 300 | N/A | N/A | N/A | N/A |
| - Other material, solvent, and maintenance | N/A | 300 | N/A | N/A | N/A | N/A |
| **Shipment Cost**** | 138 | 147 | 358 | 163 | 126 | 126 |
| **Fixed total over 5 years** | 69,000 | 53,503 | 35,082 | 68,192 | 1,695 | 126 |
| **Variable unit cost per sample** | 0.04 | 0.04 | 0.09 | 0.04 | 0.04 | 3.06 |

*Device costs are inclusive of Lao PDR VAT rate of 10%

**Shipment cost was estimated from the average price of DHL Express Worldwide service from Europe (UK) and the USA to Lao PDR based on device weight (June, 2018).

other than the PADs was the initial purchase cost. The PADs had the highest variable costs per sample, estimated at US$ 3.06 with no capital costs. For the other devices the variable cost per sample were below US$ 0.10 (Table 3).

## Cost of implementation

To implement the post-market surveillance with these devices nationally with a 1-sample strategy would incur an initial cost ranging between US$5,308 to US$2,893,292 in both prevalence scenarios (Table 4). The TruScan RM had the highest total initial cost followed by Progeny, MicroPHAZIR RX, 4500a FTIR, NIR-S-G1 and PADs, respectively. The total annual costs ranged from US$220,706 to US$380,317 under the high prevalence scenario and from US$139,925 to US$196,302 under the lower prevalence scenario. The MicroPHAZIR RX had the highest annual costs followed by the TruScan RM, 4500a FTIR, NIR-S-G1, PADs, and Progeny.

The procurement cost of devices is the largest cost category for all devices under both scenarios (Fig 2A and 2B) except NIR-S-G1 and PADs, for which the largest cost category is the cost of reference testing and the cost of inspections, respectively, under the high prevalence scenario (Fig 2A) and the cost of inspections for both devices under the lower prevalence scenario (Fig 2C). Without accounting for the cost of devices, the reference testing cost category is the largest for all devices under high prevalence scenario except for PADs for which the cost of inspections is the largest cost category (Fig 2B). The cost of inspections are the largest cost category for all devices under the lower prevalence scenario except NIR-S-G1 where the cost of reference testing is highest (Fig 2D).

## Cost-effectiveness of the devices individually compared with visual inspections alone

In the high prevalence scenario and using a single-sample strategy, the devices averted between 445 and 778 DALYs per year across the malaria endemic areas in Laos, compared with a baseline of visual inspections alone, with the MicroPHAZIR RX being the most effective device. All devices were cost-effective when compared with the baseline of visual inspections alone, with an ICER well below the WTP threshold (indicated by the blue line in Fig 3A), ranging between US$391 and US$1,514 per DALY averted.

**Table 4. Cost of implementation at country level with two prevalence scenarios.**

**a)** Cost of implementation at country level under a high prevalence scenario (20% substandard and 20% falsified)

| Cost US$ (2017) | TruScan RM | Micro PHAZIR RX | 4500a FTIR | Progeny | NIR-S-G1 | PADs |
|---|---|---|---|---|---|---|
| Initial Cost | | | | | | |
| Cost of Devices* | 2,887,500 | 2,194,500 | 1,458,414 | 2,832,855 | 64,634 | 0 |
| Shipping Cost** | 5,792 | 6,173 | 15,047 | 6,864 | 5,308 | 5,308 |
| Total Initial Cost | **2,893,292** | **2,200,673** | **1,473,461** | **2,839,719** | **69,942** | **5,308** |
| Annual Cost | | | | | | |
| Maintenance cost | 1,176 | 11,613 | - | 6,090 | 315 | - |
| Cost of Inspections§ | 81,993 | 81,984 | 82,099 | 82,072 | 81,959 | 82,290 |
| Cost of Consumablesß | 492 | 474 | 1,050 | 648 | 423 | 23,917 |
| Cost of Reference analysis by HPLC† | 183,538 | 197,656 | 169,420 | 127,065 | 164,286 | 79,062 |
| Cost of Replacement of suspected SF ACTsΣ | 82,262 | 88,590 | 75,934 | 56,950 | 73,633 | 35,436 |
| **Total Annual Cost** | **349,460** | **380,317** | **328,503** | **272,825** | **320,615** | **220,706** |

**b)** Cost of implementation at country level under a low prevalence scenario (10% substandard and 5% falsified)

| Cost US$ (2017) | TruScan RM | Micro PHAZIR RX | 4500a FTIR | Progeny | NIR-S-G1 | PADs |
|---|---|---|---|---|---|---|
| Initial Cost | | | | | | |
| Cost of Devices* | 2,887,500 | 2,194,500 | 1,458,414 | 2,832,855 | 64,634 | 0 |
| Shipping Cost** | 5,792 | 6,173 | 15,047 | 6,864 | 5,308 | 5,308 |
| Total Initial Cost | **2,893,292** | **2,200,673** | **1,473,461** | **2,839,719** | **69,942** | **5,308** |
| Annual Cost | | | | | | |
| Maintenance cost | 1,176 | 11,613 | - | 6,090 | 315 | - |
| Cost of Inspections§ | 81,993 | 81,984 | 82,099 | 82,072 | 81,959 | 82,290 |
| Cost of Consumablesß | 491 | 474 | 1,050 | 648 | 423 | 23,917 |
| Cost of Reference analysis by HPLC† | 63,532 | 70,592 | 56,473 | 35,296 | 55,190 | 28,237 |
| Cost of Replacement of suspected SF ACTsΣ | 28,475 | 31,639 | 25,311 | 15,820 | 24,736 | 12,656 |
| **Total Annual Cost** | **175,667** | **196,302** | **164,934** | **139,925** | **162,623** | **147,099** |

*Device costs are inclusive of Lao PDR VAT rate at 10%.

** Shipping cost was estimated from the average price of DHL Express Worldwide service from Europe (UK) and the USA to Lao PDR based on device weight.

§Cost of inspections was estimated based on the total time spent for overall inspections (visual inspections) and additional time spent for the test by each device.

ßCost of consumables was estimated from additional material used including reagent and cleaning wipes for the test by each device.

†Cost of reference analysis was estimated from the number of samples sent to validate with HPLC from suspected substandard and falsified (SF) samples as suggested by the device screening result.

Σ Cost of replacement was estimated from cost of the whole batch of ACTs that require replacement with genuine ACTs at the pharmacy outlet due to a suspected SF batch as indicated by the device screening results.

In the lower prevalence scenario, the devices averted between 111 and 278 DALYs with a single-sample strategy, compared with the baseline of visual inspections alone; the MicroPHAZIR RX remained the most effective device. Four devices, the NIR-S-G1, PADs, 4500a FTIR, and MicroPHAZIR RX were cost-effective in this scenario with an ICER below the WTP threshold (Fig 3B) and a positive NMB (Table 5).

## Multiway comparison of all devices and sampling strategies

When comparing all possible options (six devices with 1/2/3-sample strategy) with visual inspections in Scenario 1, all options were cost-effective (Fig 4A). The comparative cost-effectiveness analysis suggests that the best option may be the NIR-S-G1 with a 2-sample strategy followed by a 3- and 1-sample strategy (Table 6). In most cases a 2-sample strategy

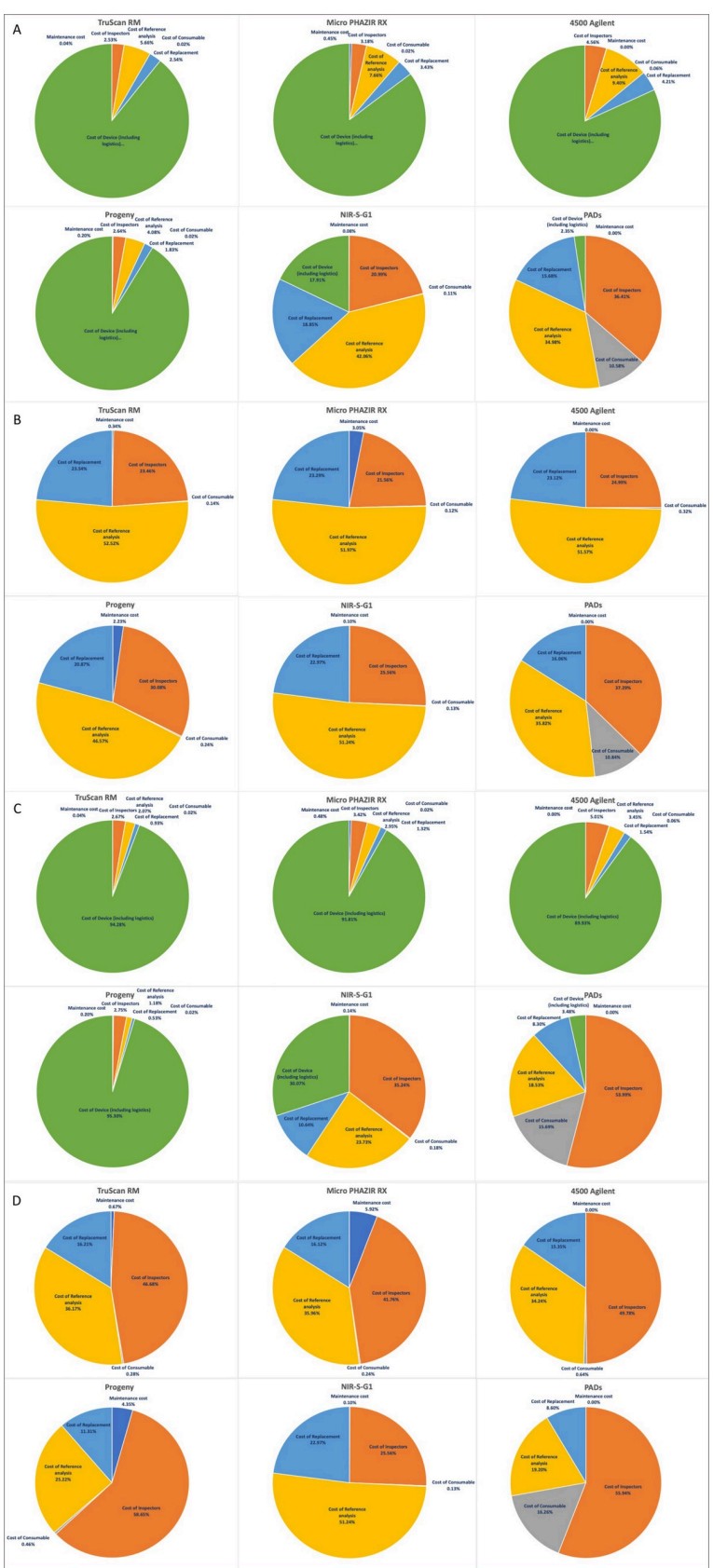

**Fig 2.** Cost of introducing post-market surveillance at country level by cost category under (A) Cost of introduction by component under a high prevalence scenario, (B) Cost of introduction by component excluding procurement costs under a high prevalence scenario, (C) Cost of introduction by component under the lower prevalence scenario, (D) Cost of introduction by component excluding procurement costs under the lower prevalence scenario.

outperformed the 3-sample and single sample strategies (except in the case of PADs and Progeny where a single sample strategy outperformed 2/3-sample strategies).

In Scenario 2, when comparing all possible options (six devices and 1/2/3-sample strategy) with visual inspections, 12 out of 18 options were cost-effective (Fig 4B). The comparative cost-effectiveness analysis suggests that in this scenario too the best option would be using NIR-S-G1 with 2-samples, followed by 3-samples, and 1-sample (Table 6).

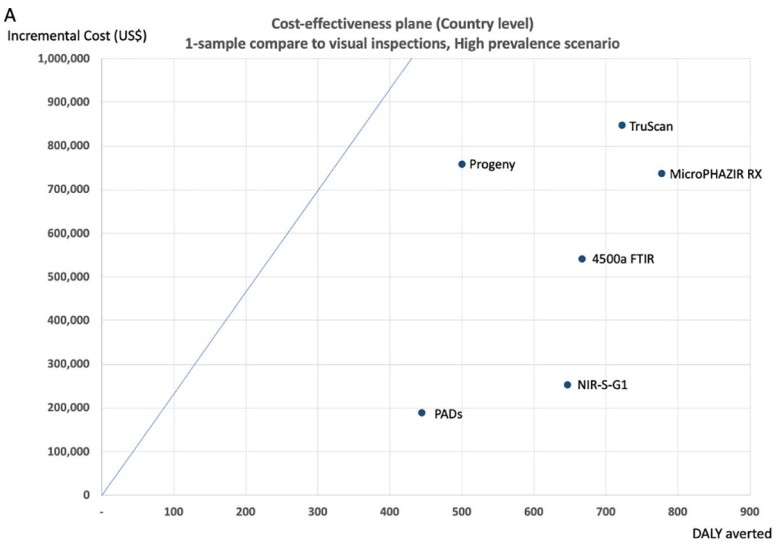

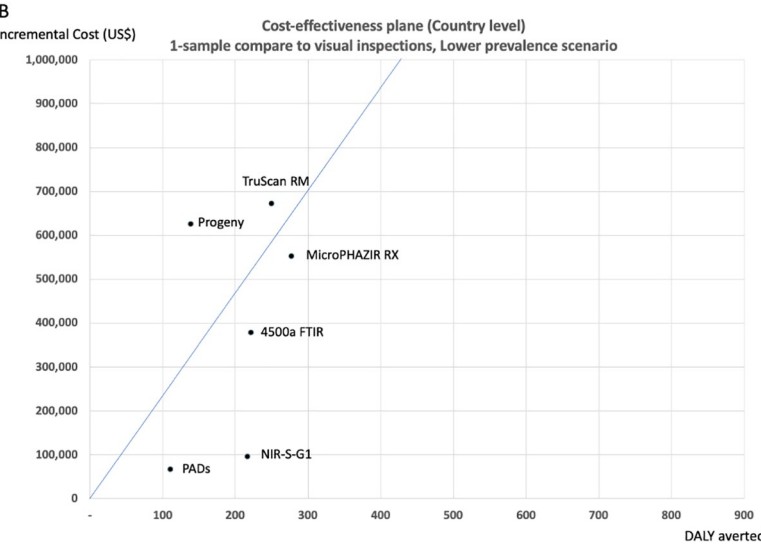

**Fig 3.** Incremental costs and effects of inspection with a 1-sample strategy in (A) high prevalence scenario; 20% substandard and 20% falsified and in (B) lower prevalence scenario; substandard 10% and falsified 5%, compared with visual inspection. The diagonal line represents the Willingness to pay threshold at US$ 2,353, Lao PDR GDP per capita).

**Table 5. Country level costs and effects in high and lower prevalence scenario per device with a 1-sample strategy compared with visual inspection ranked by descending net monetary benefit (NMB, US$).** DALYs—disability adjusted life years. ICER—incremental cost-effectiveness ratio.

| Device name | Cost | DALYs | Incremental Cost | DALY averted | ICER | NMB |
|---|---|---|---|---|---|---|
| **High prevalence scenario** | | | | | | |
| Baseline | 81,900 | 1,111.7 | | | | |
| NIR-S-G1 | 334,541 | 464.9 | 252,641 | 647 | 391 | 1,269,333 |
| MicroPHAZIR RX | 818,129 | 333.5 | 736,229 | 778 | 946 | 1,094,896 |
| 4500a FTIR | 623,195 | 444.7 | 541,295 | 667 | 811 | 1,028,241 |
| PADs | 270,838 | 667.0 | 188,938 | 445 | 425 | 857,419 |
| TruScan RM | 927,883 | 389.1 | 845,983 | 723 | 1,171 | 854,348 |
| Progeny | 839,551 | 611.4 | 757,651 | 500 | 1,514 | 419,501 |
| **Low prevalence scenario** | | | | | | |
| Baseline | 81,900 | 444.7 | | | | |
| NIR-S-G1 | 176,548 | 227.4 | 94,648 | 217 | 436 | 416,640 |
| PADs | 148,161 | 333.5 | 66,261 | 111 | 596 | 195,328 |
| 4500a FTIR | 459,626 | 222.3 | 377,726 | 222 | 1,699 | 145,452 |
| MicroPHAZIR RX | 634,114 | 166.8 | 552,214 | 278 | 1,987 | 101,759 |
| TruScan RM | 754,091 | 194.6 | 672,191 | 250 | 2,687 | -83,615 |
| Progeny | 706,651 | 305.7 | 624,751 | 139 | 4,496 | -297,765 |

## Sensitivity analysis

Fig 5 shows a tornado diagram for the case of the NIR-S-G1 under Scenario 2, illustrating the change in NMB when key model parameters were changed above or below their point estimates for the NIR-S-G1. The number of months after detecting a SF batch of ACTs before returning to baseline SF prevalence has the highest impact on the NMB, followed by the device performance in detecting genuine ACTs. Results of the one-way sensitivity analysis for all other devices in the lower prevalence scenario are provided in Fig A in S1 Text. With lower detection rates of visual inspection in both scenarios, all but one of the devices (the Progeny) were cost-effective with lower ICERs in both high and lower prevalence scenarios (Table B in S1 Text).

When the total number of devices across the country is reduced to 5 (from 42), all devices, and especially devices with high upfront costs such as the 4500a FTIR, MicroPHAZIR RX, TruScan RM, and Progeny were highly cost-effective. In this scenario, MicroPHAZIR RX was estimated to be the most cost-effective option (Table 7 and Table C in S1 Text).

## Budget impact analysis

The total budget to implement the routine medicine inspections with these screening devices with a 1-sample strategy across all 42 districts over five years ranged from US$1,108,836 to US$4,640,594 under high prevalence scenario and from US$704,806 to US$3,771,629 under lower prevalence scenario. TruScan RM was associated with the highest 5-year total cost followed by Progeny, MicroPHAZIR RX, 4500a FTIR, NIR-S-G1, and PADs, respectively. (Table 8).

## Discussion

Our results indicate that introduction of any of the devices included here into routine pharmacy post market surveillances for the detection of SF ACTs would be cost-effective in a scenario where SF antimalarials are highly prevalent. In a scenario where substandard medicines

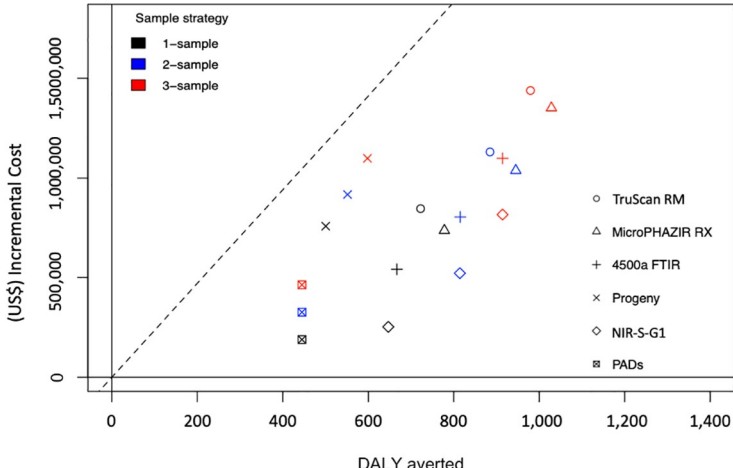

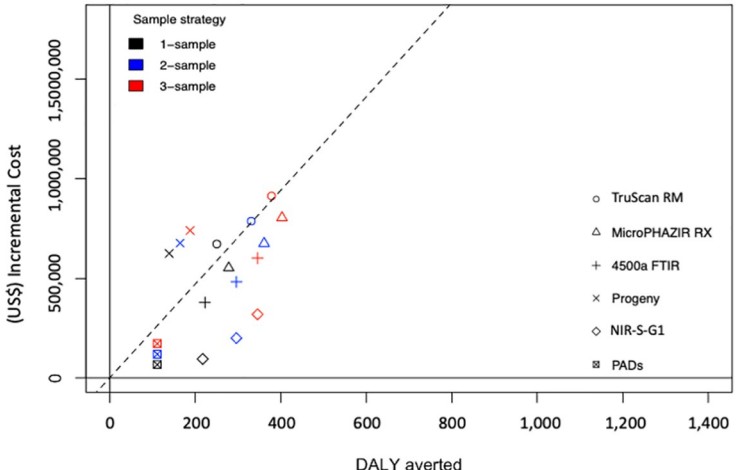

**Fig 4.** Incremental costs and effects of all sampling strategies (1, 2, or 3-samples per drug per inspection) under (A) high prevalence scenario and (B) lower prevalence scenario. The diagonal line represents the Willingness to pay threshold at US$ 2,353, Lao PDR GDP per capita.

are prevalent but falsified medicines are less frequent, there is a clear advantage for devices that can detect both forms of poor quality medicines. In both scenarios, the NIR-S-G1 appeared to be the most cost-effective option, with a 2-sample strategy being more cost-effective compared with 3- and 1- sample tests per brand of ACT, with the incremental costs of the 3$^{rd}$ sample as compared with the first 2 samples not being justified by the incremental gains. The MicroPHAZIR was associated with the best performance in identifying SF antimalarials, but due to its high initial costs as compared with the variable costs, it was the most cost-effective option only when fewer devices are procured for use across several districts, which could be logistically impractical. The budget impact analysis also suggests that some of these devices could be affordable for local authorities, in particular the 4500 Agilent, NIR-5-G1, and PADs. Our findings show that for the majority of devices, the initial procurement cost alone is much higher than the annual cost of visual ACT inspection alone (ranging from 18 times higher with the 4500 Agilent to 35 times higher with TruScan RM), with the exception of PADs and NIR-

**Table 6. Country level costs and effects in high and lower prevalence scenario per device and per different sample strategy compared with visual inspection ranked by descending net monetary benefits (NMB, US$).** DALY—disability adjusted life year. ICER—incremental cost-effectiveness ratio.

| High prevalence scenario | | | | | | |
|---|---|---|---|---|---|---|
| Rank | Device | Strategy | Incremental Cost | DALY averted | ICER | NMB |
| 1 | NIR-S-G1 | 2-sample | 521,578 | 814.4 | 640 | 1,394,582 |
| 2 | NIR-S-G1 | 3-sample | 816,210 | 914 | 893 | 1,334,537 |
| 3 | NIR-S-G1 | 1-sample | 252,641 | 646.8 | 391 | 1,269,333 |
| 4 | MicroPHAZIR RX | 2-sample | 1,038,137 | 945 | 1,099 | 1,185,372 |
| 5 | 4500a FTIR | 2-sample | 804,136 | 815.3 | 986 | 1,114,185 |
| 6 | MicroPHAZIR RX | 1-sample | 736,229 | 778.2 | 946 | 1,094,896 |
| 7 | MicroPHAZIR RX | 3-sample | 1,351,730 | 1,028.40 | 1,314 | 1,067,970 |
| 8 | 4500a FTIR | 3-sample | 1,099,001 | 914.1 | 1,202 | 1,051,844 |
| 9 | 4500a FTIR | 1-sample | 541,295 | 667 | 811 | 1,028,241 |
| 10 | TruScan RM | 2-sample | 1,130,917 | 884.8 | 1,278 | 950,897 |
| 11 | TruScan RM | 3-sample | 1,439,046 | 979.3 | 1,469 | 865,301 |
| 12 | PADs | 1-sample | 188,938 | 444.7 | 425 | 857,419 |
| 13 | TruScan RM | 1-sample | 845,983 | 722.6 | 1,171 | 854,348 |
| 14 | PADs | 2-sample | 326,192 | 444.7 | 734 | 720,166 |
| 15 | PADs | 3-sample | 463,445 | 444.7 | 1,042 | 582,912 |
| 16 | Progeny | 1-sample | 757,651 | 500.3 | 1,514 | 419,501 |
| 17 | Progeny | 2-sample | 917,220 | 551.2 | 1,664 | 379,827 |
| 18 | Progeny | 3-sample | 1,098,954 | 597.9 | 1,838 | 307,997 |
| Low prevalence scenario | | | | | | |
| Rank | Device | Strategy | Incremental Cost | DALY averted | ICER | NMB |
| 1 | NIR-S-G1 | 2-sample | 199,405 | 296.2 | 673 | 497,626 |
| 2 | NIR-S-G1 | 3-sample | 318,591 | 345.9 | 921 | 495,217 |
| 3 | NIR-S-G1 | 1-sample | 94,648 | 217.3 | 436 | 416,640 |
| 4 | 4500a FTIR | 2-sample | 481,535 | 296.5 | 1,624 | 216,037 |
| 5 | 4500a FTIR | 3-sample | 601,355 | 345.9 | 1,739 | 212,478 |
| 6 | PADs | 1-sample | 66,261 | 111.2 | 596 | 195,328 |
| 7 | MicroPHAZIR RX | 2-sample | 675,210 | 361.3 | 1,869 | 174,955 |
| 8 | 4500a FTIR | 1-sample | 377,726 | 222.3 | 1,699 | 145,452 |
| 9 | MicroPHAZIR RX | 3-sample | 804,050 | 403 | 1,995 | 144,211 |
| 10 | PADs | 2-sample | 118,805 | 111.2 | 1,069 | 142,784 |
| 11 | MicroPHAZIR RX | 1-sample | 552,214 | 277.9 | 1,987 | 101,759 |
| 12 | PADs | 3-sample | 171,349 | 111.2 | 1,541 | 90,240 |
| 13 | TruScan RM | 2-sample | 786,713 | 331.2 | 2,375 | -7,395 |
| 14 | TruScan RM | 3-sample | 912,833 | 378.5 | 2,412 | -22,249 |
| 15 | TruScan RM | 1-sample | 672,191 | 250.1 | 2,687 | -83,615 |
| 16 | Progeny | 2-sample | 676,709 | 164.4 | 4,115 | -289,775 |
| 17 | Progeny | 1-sample | 624,751 | 139 | 4,496 | -297,765 |
| 18 | Progeny | 3-sample | 739,749 | 187.8 | 3,939 | -297,863 |

5-G1where the costs of device are lower. This could act as a deterrent from their adoption, despite being potentially highly cost-effective.

While in many of the scenarios the devices appeared to be cost-effective, it is important to emphasize that cost-effectiveness is to a great extent contextual and will depend on how the devices are implemented and how their use alters pre-existing practices. For example, it was generally the case that devices with the ability to detect substandard medicines as well as

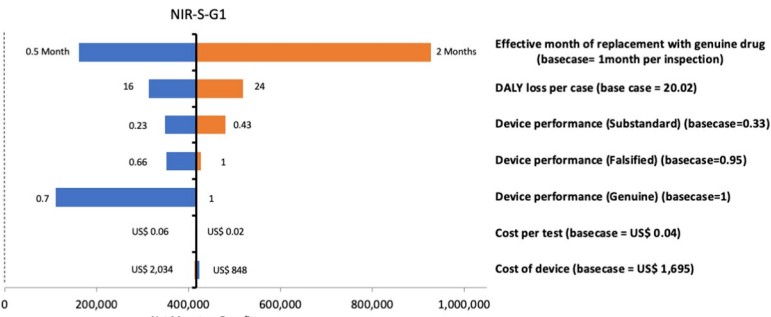

**Fig 5.** Tornado diagram illustrating one-way sensitivity analyses with a range of plausible parameter values for key model parameters, in a lower prevalence scenario for the NIR-S-G1 device (blue bars represents scenarios with lower parameter estimates, orange bars represent scenarios with the higher parameter estimates, and dotted vertical line represents the cost-effectiveness threshold).

falsified ones outperformed those that could detect falsified medicines alone; how big an advantage this represents in a real-life setting will depend mostly on the relative prevalence of substandard and falsified medicines. In a context with a large percentage of substandard medicines but lower falsified ones, only those devices with the capacity to detect substandard medicines are likely to be cost-effective. As it is increasingly apparent that medicine quality is highly variable through time and space, conclusions on cost-effectiveness will also be dynamic as the prevalence of SF medicines change and as regulatory systems, pharmaceutical supply chains, and medicine use patterns change.

The cost-effectiveness results were highly dependent on the assumptions made on how the devices would be integrated within the medicine inspection environment. For instance, the number of devices required per province and how inspectors would respond to samples that fail a test. We assumed that for logistical reasons each district requires its own device, and that when a substandard or falsified medicine is detected, the batch of medicine is replaced with a genuine one lasting for one month before returning to baseline levels of SF medicines. The detection of SF medicines may have much wider effects for public health and cost-effectiveness if the information is shared appropriately with other neighboring districts, provinces and countries which would be alerted to the problem, facilitating further investigation and response. All these factors will vary considerably through time and space, therefore at a later stage if a decision is made to proceed with introducing these devices it will be imperative to refine the assumptions and parameter estimates. A refined analysis will then be more informative as to the choice of device, and how best to utilise it in the field. The approaches and parameter estimates used in the cost-effectiveness analysis were mostly conservative and our results,

**Table 7. Country level costs and effects in sensitivity analysis when using one device per province instead of one per district per device in lower prevalence scenario with a 1-sample strategy compared with visual inspection in descending order of net monetary benefit (NMB, US$).** DALY—disability adjusted life year. ICER—incremental cost-effectiveness ratio.

| Name | Cost | DALYs | Incremental Cost | DALY averted | ICER | NMB |
|---|---|---|---|---|---|---|
| Baseline | 81,900 | 444.7 | | | | |
| MicroPHAZIR RX | 238,192 | 166.8 | 156,292 | 277.9 | 562 | 497,681 |
| NIR-S-G1 | 164,003 | 227.4 | 82,103 | 217.3 | 378 | 429,185 |
| TruScan RM | 243,491 | 194.6 | 161,591 | 250.1 | 646 | 426,985 |
| 4500a FTIR | 200,016 | 222.3 | 118,116 | 222.3 | 531 | 405,062 |
| Progeny | 202,028 | 305.7 | 120,128 | 139.0 | 864 | 206,859 |
| PADs | 147,226 | 333.5 | 65,326 | 111.2 | 588 | 196,263 |

**Table 8. Budget impact analysis under high and low prevalence scenario with a 1-sample strategy across all 42 districts.**

| Cost US$ (2017) | TruScan RM | Micro PHAZIR RX | 4500a FTIR | Progeny | NIR-S-G1 | PADs |
|---|---|---|---|---|---|---|
| High prevalence scenario | | | | | | |
| Cost for the 1st year | 3,242,752 | 2,580,990 | 1,801,964 | 3,112,544 | 390,558 | 226,014 |
| Cost for subsequent years (2nd-5th year) | 349,460 | 380,317 | 328,503 | 272,825 | 320,615 | 220,706 |
| Total cost (five years) | **4,640,594** | **4,102,258** | **3,115,976** | **4,203,845** | **1,673,019** | **1,108,836** |
| Low prevalence scenario | | | | | | |
| Cost for the 1st year | 3,068,959 | 2,396,975 | 1,638,395 | 2,979,644 | 232,565 | 152,408 |
| Cost for subsequent years (2nd-5th year) | 175,667 | 196,302 | 164,934 | 139,925 | 162,623 | 147,099 |
| Total cost (five years) | **3,771,629** | **3,182,183** | **2,298,131** | **3,539,346** | **883,055** | **740,806** |

therefore, are likely to underestimate rather than overestimate cost-effectiveness. Most importantly, we focused only on the outcomes of detecting SF ACTs and ignored other essential medicines, particularly antimicrobial agents such as amoxicillin, acyclovir, doxycycline, ethambutol and isoniazid that can cure many widely prevalent infections, and ensuing genuine medicines are used can help slow the development of antimicrobial resistant to these agents in the long run. [29,30] For some devices for which the reagents are costly and the range of API detection is limited, this was appropriate. Other devices able to detect a broad range of medicines at no added cost will offer greater potential health outcomes than those accounted for in our analysis.

The costs per test for each device were derived from capital purchase costs of the device, reagents and other consumable costs that are dependent on the number of samples tested, and maintenance costs that are mostly fixed (although are likely to rise with the number of tests performed). We assumed that the devices are used relatively infrequently, up to 180 samples per device per year, across a district's 10 pharmacies. For some devices such as the PADs or those with high reagent costs, the cost per sample tested will scale with the frequency of testing. Other devices with high purchase costs but low consumable costs could be far more cost-effective than they appeared in the analysis if used more frequently than we assumed. It is important; however, not to overlook the limit on the number of sample tests that can be performed in a single inspection, and the opportunity costs of using up inspection time that could be dedicated to other activities (e.g. visual inspections of larger volumes of samples).

The number of previous studies estimating the burden and economic impact of poor quality antimalarials in low- and middle-income countries is very limited [31–33] and only one of these assessed the economic impact of interventions that might address this impact, focusing on eliminating antimalarial stock-outs, increasing care seeking rates for malaria treatment, and encouraging better adherence to antimalarial treatment [27]. The annual loss due to SF antimalarials was estimated to be US$ 892 million in Nigeria [29] and at US$ 614 million for the population aged under five in Uganda [31]. Our work is one of the first attempts to evaluate the cost-effectiveness of interventions to mitigate the impact of SF ACTs (and SF medicines more broadly) and to our knowledge the first to consider the cost-effectiveness of devices to detect medicine quality in field inspections. We based our analysis in the context of Laos, but our findings might be informative for other countries where malaria endemicity and the prevalence of poor quality medicine in considering these devices for post-market surveillance.

This study has several limitations. First, our analysis contains many assumptions about how devices might be utilised in the inspections and about contextual factors such as the total number of ACT brands and stock level at the pharmacy. This may not be accurate and is very context specific. We used imagined prevalences of SF ACTs that are unlikely to reflect reality. In addition, the scope of this study considers only the costs and outcomes if devices are deployed

at final pharmacy points, rather than higher up the pharmaceutical supply chain. Evaluations with wider scope would help with better understanding of the broader usage and impact of these devices. Possible indirect effects of using these devices in field inspections were not accounted for, notably higher vigilance on the part of retailers in their medicine purchasing and storage practices. Lower prevalence of SF ACTs could also slow the spread of artemisinin resistance with long term health and economic gains, which are unaccounted for here. The probabilities used to measure the effectiveness of the devices was based on a study performed in the laboratory evaluation. Devices were tested 'out-of-the-box' but their performances could have been improved if further upfront work had been performed [20,21]. This may have induced an under-estimation of the probabilities to identify SF (especially substandard medicines) in the current study. The upfront work required would also have had increased their costs. Finally, we were surprised to discover a very meagre evidence base in the Greater Mekong subregion of the economics of post-market surveillance of medical products. More data, collected in standardized formats across the subregion, are needed to allow more objective comparative analysis of the economics of post-market surveillance and how such devices and other interventions would vary in cost-effectiveness in different communities.

In conclusion, our findings establish that routine use of field detection devices in post marketing surveillance in a resource limited setting could be cost-effective, but further evaluation will be needed to identify optimal strategies for their use. This information can aid policy-makers or regulators considering investment in handheld screening devices to improve medicine quality and reduce the undesired health and economic burden associated with poor quality medicines.

## Supporting information

**S1 Text. Additional information for methods and results.** Text A. List of information acquired from manufacturers for device cost estimation. Table A. Median total time taken per sample in sample set testing. Fig A. Results of sensitivity analyses from the cost-effectiveness analysis. Table B. Results from sensitivity analyses with lower detection rate with visual inspection (A) 12.5% and 25% reduction for substandard and falsified medicines, respectively and (B) 5% and 10% reduction under both high and lower prevalence scenarios. Table C. Results from a sensitivity analysis when using one device per province instead of one per district per device in high prevalence scenario with a 1-sample strategy compared with visual inspection. (DOCX)

## Acknowledgments

We are very grateful to the directors of Mahosot Hospital, the Director, Dr Manivanh Vongsouvath, and staff of the Microbiology Laboratory, Mahosot Hospital, Vientiane, Lao PDR. We thank Dr Douglas Ball, Dr Susann Roth, Dr Sonalini Khetrapal, Dr Ruth Bird and Professor Philippe Guerin for their support and advice. We thank the staff at Bureau of Food and Drug Inspection (BFDI) for assistance with estimates of costs in the Lao PDR.

## Author Contributions

**Conceptualization:** Nantasit Luangasanatip, Panarasri Khonputsa, Céline Caillet, Serena Vickers, Paul N. Newton, Yoel Lubell.

**Data curation:** Nantasit Luangasanatip, Panarasri Khonputsa, Serena Vickers, Stephen Zambrzycki, Facundo M. Fernández, Yoel Lubell.

**Formal analysis:** Nantasit Luangasanatip, Panarasri Khonputsa, Céline Caillet, Yoel Lubell.

**Funding acquisition:** Paul N. Newton.

**Investigation:** Nantasit Luangasanatip, Céline Caillet, Stephen Zambrzycki, Facundo M. Fernández, Paul N. Newton.

**Methodology:** Nantasit Luangasanatip, Panarasri Khonputsa, Céline Caillet, Serena Vickers, Yoel Lubell.

**Project administration:** Nantasit Luangasanatip.

**Supervision:** Yoel Lubell.

**Validation:** Paul N. Newton.

**Writing – original draft:** Nantasit Luangasanatip, Panarasri Khonputsa, Céline Caillet, Yoel Lubell.

**Writing – review & editing:** Nantasit Luangasanatip, Céline Caillet, Serena Vickers, Stephen Zambrzycki, Facundo M. Fernández, Paul N. Newton, Yoel Lubell.

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
