## [Decision Letter · Decision Letter 0]

6 Feb 2021

Dear Dr Luangasanatip,

Thank you very much for submitting your manuscript "Implementation of Field Detection Devices in Medicine Quality Screening in Lao PDR – A Cost-Effectiveness Analysis" for consideration at PLOS Neglected Tropical Diseases. As with all papers reviewed by the journal, your manuscript was reviewed by members of the editorial board and by several independent reviewers. In light of the reviews (below this email), we would like to invite the resubmission of a significantly-revised version that takes into account the reviewers' comments. 

We cannot make any decision about publication until we have seen the revised manuscript and your response to the reviewers' comments. Your revised manuscript is also likely to be sent to reviewers for further evaluation.

Sincerely,

Vahid Yazdi-Feyzabadi, PhD

Deputy Editor

Vahid Yazdi-Feyzabadi

Deputy Editor

Reviewer's Responses to Questions

**Key Review Criteria Required for Acceptance?**

**Methods**

-Are the objectives of the study clearly articulated with a clear testable hypothesis stated?

-Is the study design appropriate to address the stated objectives?

-Is the population clearly described and appropriate for the hypothesis being tested?

-Is the sample size sufficient to ensure adequate power to address the hypothesis being tested?

-Were correct statistical analysis used to support conclusions?

-Are there concerns about ethical or regulatory requirements being met?

Reviewer #1: First, I think that te paper needs to more clearly present the objectives of the study rather than only saying that they are evaluating the cost-effectiveness of introducing the screening devices. They should state what the testable hypothesis is for the study, the perspective of the study, and the target audience. Also, since the study focused on testing of ACTs, it appears to be focused on antimalarials rather than other types of drugs. In that case, the title of the study should be changed to reflect the focus on antimalarials as opposed to all medicines. Iis not clear whether the findings can be generalized to other types of medicines.

It would also be helpful to describe the prevalence and incidence of malaria in Lao PDR what interventions are currently being implemented against malaria, etc.

The study design is described but focuses largely on the assumptions of the decision analysis and effectiveness of the screening devices. It requires more detail on data collection and costing methods. It is unclear whether actual data collection was conducted in Lao PDR and, if so, who conducted the analysis. Also, it does not include a description of the cost categories and costing methods used. It does not appear that capital costs were depreciated as would be expected. Also, the total costs of introducing the devices is not described, i.e. implementation costs as well as the costs of the devices, are not presented except in Table 7. It would be good to see a table with total costs as well as proportions spent on each cost category since delivery costs are important in making decisions on introduction of a new technology.

I found Table 1 difficult to interpret. It would help to explain this table in more detail - e.g. what device fail and device pass means. This information is in the decision analytic model but should also be stated in the table.

Reviewer #2: Yes, the methods used, the statistical analysis and data presentation is well indicated. The study met its objectives with sufficient sample size.

Reviewer #3: Use Strategies instead of overview heading.

In the parameter inputs section of the methods, it is better to include subtitles such as probabilities, costs and consequences, which in each section, how to calculate each item is explained in detail.

Given that the measure of outcome is DALY averted in this study and the main parameters of DALY such as YLL and disability weight are assumed and there is no specific source or reference for them. Therefore, the results cannot be trusted.

The perspective of cost calculating, time horizon and discount rate are not mentioned in the method.

**Results**

-Does the analysis presented match the analysis plan?

-Are the results clearly and completely presented?

-Are the figures (Tables, Images) of sufficient quality for clarity?

Reviewer #1: The analysis presented matches the analysis plan to some extent. However, as noted above, it does not provide sufficient information on the costs of implementation and only presents this information in the budget impact. This information should be presented in the general results section for the two scenarios and some charts developed with the shares of total costs for each cost category with and without the costs of the devices

Reviewer #2: Yes

Reviewer #3: The ACER column in Tables 4 and 5 should be changed to ICER because ACER is calculated for a strategy while the numbers in this column are incremental cost/ incremental effect for a strategy compare with baseline.

Probability sensitivity analysis(PSA) should be performed according to its importance in policy decisions

**Conclusions**

-Are the conclusions supported by the data presented?

-Are the limitations of analysis clearly described?

-Do the authors discuss how these data can be helpful to advance our understanding of the topic under study?

-Is public health relevance addressed?

Reviewer #1: In the discussion section, it would be helpful to compare the findings with other studies in the literature if there are any. Also, it would be good to compare the delivery costs with and without the cost of devices with current expenditures on ACT screening if this information is available.

Reviewer #2: Discussion is poorly written.

Reviewer #3: (No Response)

**Editorial and Data Presentation Modifications?**

Reviewer #1: (No Response)

Reviewer #2: Minor revision

Reviewer #3: (No Response)

**Summary and General Comments**

Reviewer #1: The paper discusses the cost-effectiveness of of portable screening devices to assess the quality of medicines in Lao PDR. The paper indicates that insufficient screening of medicine quality is being conducted, particularly to assess substandard medicines. However, the paper needs to improve its situation analysis, presentation of methods and results, as well as discuss other literature on the subject. It is unclear whether the study is focusing on antimalarials rather than all medicines, the title does not seem to match the content, and the costing methods are not well-described and presented.

Reviewer #2: The authors sought to determine the Cost-Effectiveness of Field Detection Devices in Medicine Quality Screening in Lao PDR. Indeed, Substandard and falsified medicines pose a significant threat to the health system nowadays.

The authors did an outstanding work which add knowledge to the scientific community.

The methods used, the statistical analysis and data presentation is well indicated. The English write-up was also good.

It is a breakthrough to assess the cost-effectiveness of intervention methods to mitigate the impact of SF ACTs.

Listed below are minor comments:

• Your study focused on market surveillance among pharmacies in Laos. The study could have been more plausible if it is multi-centered including various facilities to draw better concluding remarks.

• You have tested about 6 different devices to evaluate cost-effectiveness; however, device specific conclusions were not made lastly.

• The discussion is poorly written; It looks simply a repetition of your methodology and result part. Moreover, it states facts and I didn’t see comparisons with other similar studies. Suggestion: better to mention a portion of your result, compare with other similar (may not be exactly same) reports, then draw justification.

Reviewer #3: In the abstract, instead of the analysis scenario, it is better to emphasize the type of costs that have been calculated and the outcome measurement index.

Depending on the type of economic evaluation study (cost utility analysis), it is better to use the term outcome or effect instead of benefit in all parts of the manuscript.

PLOS authors have the option to publish the peer review history of their article (what does this mean?). If published, this will include your full peer review and any attached files.

Reviewer #1: No

Reviewer #2: Yes: Melaku Ashagrie Belete

Reviewer #3: Yes: Reza Goudarzi
---

## [Decision Letter · Decision Letter 1]

2 May 2021

Dear Dr Luangasanatip,

Thank you very much for submitting your manuscript "Implementation of Field Detection Devices for Antimalarial Quality Screening in Lao PDR – A Cost-Effectiveness Analysis" for consideration at PLOS Neglected Tropical Diseases. As with all papers reviewed by the journal, your manuscript was reviewed by members of the editorial board and by several independent reviewers. The reviewers appreciated the attention to an important topic. Based on the reviews, we are likely to accept this manuscript for publication, providing that you modify the manuscript according to the review recommendations. 

Please prepare and submit your revised manuscript within 10 days. If you anticipate any delay, please let us know the expected resubmission date by replying to this email. 

Sincerely,

Vahid Yazdi-Feyzabadi, PhD

Deputy Editor

Vahid Yazdi-Feyzabadi

Deputy Editor

Reviewer's Responses to Questions

**Key Review Criteria Required for Acceptance?**

**Methods**

-Are the objectives of the study clearly articulated with a clear testable hypothesis stated?

-Is the study design appropriate to address the stated objectives?

-Is the population clearly described and appropriate for the hypothesis being tested?

-Is the sample size sufficient to ensure adequate power to address the hypothesis being tested?

-Were correct statistical analysis used to support conclusions?

-Are there concerns about ethical or regulatory requirements being met?

Reviewer #1: The objectives of the study are clearly stated. The study design seems appropriate. In general, the methods seem appropriate. I question, though, why the study authors did not use discounting for their capital costs. In a cost-effectiveness study, it is recommended to estimate economic costs rather than financial costs and the capital costs should be discounted. So, I suggest that the authors should discount the capital costs.

Reviewer #3: (No Response)

**Results**

-Does the analysis presented match the analysis plan?

-Are the results clearly and completely presented?

-Are the figures (Tables, Images) of sufficient quality for clarity?

Reviewer #1: (No Response)

Reviewer #3: (No Response)

**Conclusions**

-Are the conclusions supported by the data presented?

-Are the limitations of analysis clearly described?

-Do the authors discuss how these data can be helpful to advance our understanding of the topic under study?

-Is public health relevance addressed?

Reviewer #1: I have some comments about the conclusions. First, for the budget impact analysis, I don't think it makes sense to say that donor support to purchase some of the devices would be required since that would not be supporting sustainability. Instead, it would be better to say what the most affordable options are, given the available budget. 

It seems obvious that cost-effectiveness is not an 'inherent' feature of the devices since cost-effectiveness will vary with the probability of SF, the costs, and effectiveness. So I would reword that sentence.

I also think that it would be useful to discuss whether there are any other uses of the devices that would increase their cost-effectiveness.

Reviewer #3: (No Response)

**Editorial and Data Presentation Modifications?**

Reviewer #1: I see that the authors added a paragraph on malaria incidence to the paper. However, I think that it should be in the introduction section rather than the methods section. Also, they should say 'incidence' rather than 'incidences.' I also think that they should add a sentence on the burden of malaria illness compared to other diseases and why it would be important to ensure that ACTs are of high quality and not SF.

The phrasing of some sentences is awkward. In several places, it would be preferable to use the term 'introducing' rather than 'implementing. - e.g. pg. 5, line 123.

Reviewer #3: (No Response)

**Summary and General Comments**

Reviewer #1: In general, the revised paper has improved.

Reviewer #3: (No Response)

PLOS authors have the option to publish the peer review history of their article (what does this mean?). If published, this will include your full peer review and any attached files.

Reviewer #1: No

Reviewer #3: Yes: Reza Goudarzi

Figure Files:

Data Requirements:

Reproducibility:

References

---

## [Editor Report · Decision Letter 2]

14 May 2021

Dear Dr Luangasanatip,

Thank you very much for submitting your manuscript "Implementation of Field Detection Devices for Antimalarial Quality Screening in Lao PDR – A Cost-Effectiveness Analysis" for consideration at PLOS Neglected Tropical Diseases. As with all papers reviewed by the journal, your manuscript was reviewed by members of the editorial board and by several independent reviewers. The reviewers appreciated the attention to an important topic. Based on the reviews, we are likely to accept this manuscript for publication, providing that you modify the manuscript according to the review recommendations. 

Please prepare and submit your revised manuscript within 7 days. If you anticipate any delay, please let us know the expected resubmission date by replying to this email. 

Sincerely,

Vahid Yazdi-Feyzabadi, PhD

Deputy Editor

Vahid Yazdi-Feyzabadi

Deputy Editor

Figure Files:

Data Requirements:

Reproducibility:

References

---

## [Editor Report · Decision Letter 3]

28 May 2021

Dear Dr Luangasanatip,

Thank you very much for submitting your manuscript "Implementation of Field Detection Devices for Antimalarial Quality Screening in Lao PDR – A Cost-Effectiveness Analysis" for consideration at PLOS Neglected Tropical Diseases. As with all papers reviewed by the journal, your manuscript was reviewed by members of the editorial board and by several independent reviewers. The reviewers appreciated the attention to an important topic. Based on the reviews, we are likely to accept this manuscript for publication, providing that you modify the manuscript according to the review recommendations. 

We did receive no response or any required change for below comments in last review. 

Please kindly response to these comments and amend based on the reviewer comments. 

1. I see that the authors added a paragraph on malaria incidence to the paper. However, I think that it should be in the introduction section rather than the methods section. Also, they should say 'incidence' rather than 'incidences.' I also think that they should add a sentence on the burden of malaria illness compared to other diseases and why it would be important to ensure that ACTs are of high quality and not SF.

2. The phrasing of some sentences is awkward. In several places, it would be preferable to use the term 'introducing' rather than 'implementing. - e.g. pg. 5, line 123.

Editor's Comments to Author

Sincerely,

Vahid Yazdi-Feyzabadi, PhD

Deputy Editor

Vahid Yazdi-Feyzabadi

Deputy Editor

We did receive no response or required change to below comments in last review. 

Please kindly response to these comments and amend based on the reviewer comments. 

I see that the authors added a paragraph on malaria incidence to the paper. However, I think that it should be in the introduction section rather than the methods section. Also, they should say 'incidence' rather than 'incidences.' I also think that they should add a sentence on the burden of malaria illness compared to other diseases and why it would be important to ensure that ACTs are of high quality and not SF.

The phrasing of some sentences is awkward. In several places, it would be preferable to use the term 'introducing' rather than 'implementing. - e.g. pg. 5, line 123.

Editor's Comments to Author

Figure Files:

Data Requirements:

Reproducibility:

References

---

## [Editor Report · Decision Letter 4]

4 Jun 2021

Dear Dr Luangasanatip,

We are pleased to inform you that your manuscript 'Implementation of Field Detection Devices for Antimalarial Quality Screening in Lao PDR – A Cost-Effectiveness Analysis' has been provisionally accepted for publication in PLOS Neglected Tropical Diseases.

Best regards,

Vahid Yazdi-Feyzabadi, PhD

Deputy Editor

Vahid Yazdi-Feyzabadi

Deputy Editor

---

## [Editor Report · Acceptance letter]

23 Aug 2021

Dear Dr Luangasanatip,

We are delighted to inform you that your manuscript, "Implementation of Field Detection Devices for Antimalarial Quality Screening in Lao PDR – A Cost-Effectiveness Analysis," has been formally accepted for publication in PLOS Neglected Tropical Diseases.

Best regards,

Shaden Kamhawi

co-Editor-in-Chief

Paul Brindley

co-Editor-in-Chief
